# Heterogeneity and Risk of Bias in Studies Examining Risk Factors for Severe Illness and Death in COVID-19: A Systematic Review and Meta-Analysis

**DOI:** 10.3390/pathogens11050563

**Published:** 2022-05-10

**Authors:** Abraham Degarege, Zaeema Naveed, Josiane Kabayundo, David Brett-Major

**Affiliations:** Department of Epidemiology, College of Public Health, University of Nebraska Medical Center, Omaha, NE 68198, USA; zaeema_arif@hotmail.com (Z.N.); josiane.kabayundo@unmc.edu (J.K.); david.brettmajor@unmc.edu (D.B.-M.)

**Keywords:** COVID-19, severe illness, death, demographics, comorbidities

## Abstract

This systematic review and meta-analysis synthesized the evidence on the impacts of demographics and comorbidities on the clinical outcomes of COVID-19, as well as the sources of the heterogeneity and publication bias of the relevant studies. Two authors independently searched the literature from PubMed, Embase, Cochrane library, and CINAHL on 18 May 2021; removed duplicates; screened the titles, abstracts, and full texts by using criteria; and extracted data from the eligible articles. The variations among the studies were examined by using Cochrane, Q.; I^2^, and meta-regression. Out of 11,975 articles that were obtained from the databases and screened, 559 studies were abstracted, and then, where appropriate, were analyzed by meta-analysis (*n* = 542). COVID-19-related severe illness, admission to the ICU, and death were significantly correlated with comorbidities, male sex, and an age older than 60 or 65 years, although high heterogeneity was present in the pooled estimates. The study design, the study country, the sample size, and the year of publication contributed to this. There was publication bias among the studies that compared the odds of COVID-19-related deaths, severe illness, and admission to the ICU on the basis of the comorbidity status. While an older age and chronic diseases were shown to increase the risk of developing severe illness, admission to the ICU, and death among the COVID-19 patients in our analysis, a marked heterogeneity was present when linking the specific risks with the outcomes.

## 1. Introduction

While the numbers of cases and deaths during the COVID-19 pandemic are a moving target, over half a billion persons have been infected worldwide, with more than 6 million deaths as of 1 May 2022 [1]. The disease outcomes from SARS-CoV-2 infection may vary, which is due to a range of health and demographic factors [2,3,4]. Indeed, several meta-analyses studies have correlated hypertension, cardiovascular disease, an older age, and the male sex with severe illness and mortality in COVID-19 patients [5,6,7,8,9,10,11,12,13,14,15,16,17,18,19]. However, the associations with other comorbidities have been less clear, such as with cancer, chronic liver disease, diabetes, and kidney disease. While several meta-analyses studies report an increased risk of death in COVID-19 patients with diabetes [5,11,19], cancer [20,21,22], liver disease [23,24,25], and kidney disease [26,27,28,29], some report the lack of correlation between these comorbidities and the disease progression or the clinical outcomes of COVID-19 patients [30,31,32]. 

Previous meta-analyses studies have mainly been based on articles that were published in 2020 and that reflect high levels of heterogeneity and publication bias. Some of the used data were non-peer-reviewed or they originated only from one country, and some analyzed small numbers of articles. This adds to the uncertainty as to the effects of the comorbidities and the sociodemographic status on COVID-19-related outcomes. 

The relevant literature that is important to incorporate into systematic reviews on COVID-19 continues to grow as our collective experience with the disease grows. A better understanding of both the striking and subtle variations in patients, and of their circumstances and their experiences with SARS-CoV-2 infection, would inform the continued development of improved interventions to reduce the morbidity and mortality from COVID-19. In addition, knowledge of the sources of heterogeneity and publication bias guide the strategies for combining articles in meta-analyses to accurately estimate the predictors of severe illness and death in COVID-19 patients. Therefore, by following the PRISMA guidelines, we systematically summarized and assessed literature published before 18 May 2021 in order to provide updated data on the role of comorbidities in COVID-19-related clinical outcomes and deaths. Where appropriate to the data, we conducted a meta-analysis of the risk factors of severe illness, admission to the ICU, and death among COVID-19 patients, and we assessed the sources of heterogeneity and publication bias among the studies. 

## 2. Materials and Methods

### 2.1. Protocol and Registration

A protocol that was developed and that followed the Preferred Reporting Items for Systematic Reviews and Meta-analyses (PRISMA) checklist guided the execution and the reporting of this meta-analysis (Appendix A) [33]. The protocol is registered in the PROSPERO international prospective register of systematic reviews (ID = CRD42020184440) [34].

### 2.2. Inclusion and Exclusion Criteria

All retrospective, cross-sectional, and prospective clinical and epidemiological studies, except for individual case studies (comprehensive inclusion case series were allowed) that reported the prevalence or odds of death, severe illness, or admission to the ICU, and that were stratified by the comorbidities or the demographic status among COVID-19 patients, were included in the meta-analysis. Unpublished studies and non-peer-reviewed preprints in repositories, case studies with fewer than 10 samples or reports, letters, conference abstracts, protocols, gray literature, review protocols and articles, irrelevant studies (on different topics), and animal or in vitro studies were excluded. However, these sources were used to find additional eligible studies.

### 2.3. Outcome and Exposure Measures

The primary outcome was death. The secondary outcomes were severe illness and admission to the ICU. Severe illness was defined as having one of the following: an SpO2 < 94%; a respiratory rate > 30 breaths per minute; lung infiltrates in >50% of the lung fields by either plain or computed tomography radiography; an arterial partial pressure of the oxygen to fraction of the inspired oxygen (PaO2/FiO2) < 300 mmHg; or organ dysfunction. Organ dysfunction includes: respiratory failure, as evidenced by mechanical ventilation; myocardial injury, as evidenced by the need for catheterization or troponin elevation; and renal injury, as evidenced by the need for dialysis; or a 50% decrease in the renal function as assessed by either a creatinine rise or a decline in the glomerular filtration rate, hepatic failure, pulmonary embolus, or stroke/cerebrovascular accident [35].

The exposure variables were demographic (age, gender, tobacco use), and the comorbidities included hypertension, diabetes, cardiovascular disease, chronic respiratory disease, chronic kidney disease, chronic liver disease, cerebrovascular disease, and cancer. 

### 2.4. Search Methods for Identification of Studies

An article search was conducted in parallel in PubMed, Embase, Cochrane Library, and CINAHL on 6 May 2020. Articles published after 6 May 2020 in PubMed, Cochrane Library, and CINAHL were also searched on 18 May 2021. The search terms were (obesity OR hypertension OR Asthma or nutrition OR age OR gender OR ethnicity OR race OR income OR poverty OR pregnancy OR “Breastfeeding” OR “medical conditions” OR medications OR “chronic diseases” OR influenza or stroke OR HIV OR cancer OR diabetes OR “cardiovascular disease” OR “coronary heart disease” OR “chronic respiratory disease” OR “Sequential Organ Failure Assessment” OR smoking OR “co infection” OR comorbidity OR comorbidities or risk) AND (clinical OR severe OR complications OR mortality OR death) AND (“coronavirus disease 2019” OR “COVID-19” OR “Severe acute respiratory syndrome coronavirus 2” OR SARS-CoV-2 OR “Coronavirus 2” OR “2019 novel coronavirus”). Additional details of the search are available in the Appendix A. After transferring the articles that were searched from the four databases to RefWorks and after the removal of the duplicates, the titles and abstracts were screened on the basis of the inclusion and exclusion criteria. The articles that were approved for full-text review were further screened on the basis of the eligibility criteria. The article search and the screening processes were conducted by two authors independently. The two authors resolved differences by discussion. A third author was available to mediate disagreements following an independent review.

### 2.5. Data Collection

Data on the author, the study area/country, the study design, the sample size, and the crude or adjusted odds ratios (ORs) of death, severe illness, or admission to the ICU, along with a 95% confidence interval (CI) among COVID-19 patients with comorbidities vs. without comorbidities, or with different demographic statuses, were abstracted from each study. In addition, when the OR was not reported, raw data were used to estimate the crude ORs of death, with severe illness or admission to the ICU among COVID-19 patients with comorbidities vs. without comorbidities, or different demographic statuses per study. Two authors abstracted and entered the data into the excel sheet independently and compared their results. The two authors approved the final data that were used for analysis after discussion. 

### 2.6. Quality and Risk of Bias

The risk of bias and the quality of the studies that are included in this review were evaluated by using the Effective Public Health Practice Project tool [36]. The Effective Public Health Practice Project tool uses six criteria: selection bias, design, confounders, blinding, data collection, and withdrawal/dropout, in order to examine the quality of studies. Each study’s quality was determined as either low, moderate, or high for each of the six criteria by using two items for each criterion. The study’s overall quality was determined as high when the study had no weak rating for each of the six characteristics, and it was determined as moderate when the study had one weak rating in one of the six characteristics. The studies were grouped as low quality when the ratings for two or more characteristics were low. 

### 2.7. Data Analysis

Stata software version 16 was used for the data analysis (Stata Corporation, College Station, TX, USA). The percent residual variation among the studies due to heterogeneity was estimated by using Moran’s I-squared [37]. The statistical significance of the heterogeneity was tested by using Cochran’s Q chi-square test [37]. The odds ratios of the studies that were combined in the meta-analysis to estimate the summary OR were both adjusted and unadjusted estimates. A random-effect model using the Der Simonian and Laird method was used to estimate the summary ORs [38]. When the heterogeneity was high (I^2^ > 50%), a subgroup analysis was performed to estimate the summary ORs after grouping the studies by study area/country, study design, sample size, income group, and year of publication (2020 vs. 2021). Meta-regression was used to explore the sources of heterogeneity at the study-level covariates by examining the linear relationship between the ORs and the study area/country, the study design, the sample size, and the year of publication [39]. A funnel plot that displays the odds-ratio estimates against their standard errors was used to evaluate the publication bias among the studies included in the meta-analyses. The statistical significance of the asymmetry of the funnel plot was tested by using Egger’s regression test (bias if *p* < 0.1) [40]. A 95% CI and an alpha of 0.05 were used to assess the significance of the OR. 

## 3. Results

The initial search of articles from the databases on 6 May 2020 resulted in 4275 articles (PubMed: 1986; Embase: 2006; CINAHL: 224; and Cochrane Library: 59), out of which 1527 were duplicates (Figure 1). An additional search of articles published between 6 May 2020 and 18 May 2021 in PubMed (*n* = 3735), CINAHL (*n* = 2257), and Cochrane Library (*n* = 1708) yielded 7700 articles, out of which 585 were duplicates. Of the non-duplicate 9850 articles, 8274 were ineligible after screening the titles and abstracts, and 1017 articles were excluded after full-text reviews. This resulted in 559 articles that were appropriate for inclusion in the systematic review, and 542 of them were also included in the meta-analysis [3,41,42,43,44,45,46,47,48,49,50,51,52,53,54,55,56,57,58,59,60,61,62,63,64,65,66,67,68,69,70,71,72,73,74,75,76,77,78,79,80,81,82,83,84,85,86,87,88,89,90,91,92,93,94,95,96,97,98,99,100,101,102,103,104,105,106,107,108,109,110,111,112,113,114,115,116,117,118,119,120,121,122,123,124,125,126,127,128,129,130,131,132,133,134,135,136,137,138,139,140,141,142,143,144,145,146,147,148,149,150,151,152,153,154,155,156,157,158,159,160,161,162,163,164,165,166,167,168,169,170,171,172,173,174,175,176,177,178,179,180,181,182,183,184,185,186,187,188,189,190,191,192,193,194,195,196,197,198,199,200,201,202,203,204,205,206,207,208,209,210,211,212,213,214,215,216,217,218,219,220,221,222,223,224,225,226,227,228,229,230,231,232,233,234,235,236,237,238,239,240,241,242,243,244,245,246,247,248,249,250,251,252,253,254,255,256,257,258,259,260,261,262,263,264,265,266,267,268,269,270,271,272,273,274,275,276,277,278,279,280,281,282,283,284,285,286,287,288,289,290,291,292,293,294,295,296,297,298,299,300,301,302,303,304,305,306,307,308,309,310,311,312,313,314,315,316,317,318,319,320,321,322,323,324,325,326,327,328,329,330,331,332,333,334,335,336,337,338,339,340,341,342,343,344,345,346,347,348,349,350,351,352,353,354,355,356,357,358,359,360,361,362,363,364,365,366,367,368,369,370,371,372,373,374,375,376,377,378,379,380,381,382,383,384,385,386,387,388,389,390,391,392,393,394,395,396,397,398,399,400,401,402,403,404,405,406,407,408,409,410,411,412,413,414,415,416,417,418,419,420,421,422,423,424,425,426,427,428,429,430,431,432,433,434,435,436,437,438,439,440,441,442,443,444,445,446,447,448,449,450,451,452,453,454,455,456,457,458,459,460,461,462,463,464,465,466,467,468,469,470,471,472,473,474,475,476,477,478,479,480,481,482,483,484,485,486,487,488,489,490,491,492,493,494,495,496,497,498,499,500,501,502,503,504,505,506,507,508,509,510,511,512,513,514,515,516,517,518,519,520,521,522,523,524,525,526,527,528,529,530,531,532,533,534,535,536,537,538,539,540,541,542,543,544,545,546,547,548,549,550,551,552,553,554,555,556,557,558,559,560,561,562,563,564,565,566,567,568,569,570,571,572,573,574,575,576,577,578,579,580,581,582,583,584,585,586,587,588,589,590,591,592,593,594,595,596,597,598]. The majority of the studies were conducted in China, Italy, and the United States (Appendix A). The study designs were retrospective, cross-sectional, and prospective. The clinical outcomes that are reported are death, severe illness, and admission to the ICU. The exposure variables that are examined in the studies include hypertension, cardiovascular disease, diabetes, chronic respiratory disease, cancer, chronic kidney disease, chronic liver disease, cerebrovascular disease, smoking, age, and sex.

### 3.1. Hypertension

Out of the 559 studies that were approved for inclusion in this review, 302 examined the correlation between hypertension and death, severe illness, and admission to the ICU among COVID-19 patients. Appendix A and Figure 2 show the summary estimates of the odds ratios of death (vs. survival: *n* = 209 studies), severe illness (vs. moderate or mild: *n* = 100 studies), and admission to the ICU (vs. non-ICU: *n* = 36 studies) among hypertensive vs. normotensive patients, respectively. While some studies report increased odds of death (*n* = 82 studies), severe illness (*n* = 48 studies), and admission to the ICU (*n* = 11 studies) among hypertensive patients, others report the lack of correlation between hypertension and death (*n* = 122 studies), severe illness (*n* = 52 studies), or admission to the ICU (*n* = 25 studies). Few studies document lower death (*n* = 5 studies) and admission to the ICU (*n* = 1 studies) among hypertensive vs. normotensive patients. A summary analysis of the pooled data from these studies showed moderate to high heterogeneity, although there were greater odds of death (OR 1.38, 95% CI 1.30–1.46, I^2^ = 77.3%, number of studies, (*n*) = 209) (Appendix A), severe illness (OR 1.59, 95% CI 1.41–1.76, I^2^ = 47.9%, *n* = 100) (Appendix A), and admission to the ICU (OR 1.29, 95% CI 1.10–1.49, I^2^ = 63.1%, *n* = 36) among hypertensive compared to normotensive patients (Figure 2). 

### 3.2. Cardiovascular Disease

A total of 189 studies examined the nature of the relationship between cardiovascular disease and the odds of developing death (*n* = 123), severe illness (*n* = 61), and admission to the ICU (*n* = 5) among COVID-19 patients. Out of the 189 studies, 56 showed higher odds of death, and 27 showed higher odds of severe illness among COVID-19 patients with cardiovascular disease. One study showed lower odds of death among COVID-19 patients with cardiovascular disease. The remaining 66 out of 123 studies showed a lack of association between cardiovascular disease and death, and 34 out of 61 showed a lack of association between cardiovascular disease and the odds of developing severe illness among COVID-19 patients. A meta-analysis of the studies showed higher odds of death (OR 1.63, 95% CI 1.51–1.75, I^2^ = 80.2%, *n* = 123) (Appendix A) and severe illness (OR 1.27, 95% CI 1.07–1.47, I^2^ = 20.5%, *n* = 61) (Appendix A) among COVID-19 patients who had cardiovascular disease compared to those without this health problem. The odds of admission to the ICU were comparable between those who had cardiovascular disease and those without this health problem (OR 1.18, 95% CI 0.97–1.39, I^2^ = 48.5%, *n* = 30) (Figure 3).

### 3.3. Diabetes

Diabetes is also posited to be linked to the risk of developing severe illness and death among COVID-19 patients. A total of 224 studies examined whether having diabetes is correlated with the odds of death among COVID-19 patients, 95 of which reported increased odds of death in COVID-19 patients with diabetes. The correlations between having diabetes and the odds of developing severe illness and the odds of admission to the ICU were assessed in 96 (44 reported increased odds) and 46 (16 reported increased odds) studies, respectively. A summary analysis of these studies showed greater odds of death (OR 1.56, 95% CI 1.45–1.67, I^2^ = 95.5%, *n* = 224) (Appendix A), severe illness (OR 1.51, 95% CI 1.36–1.67, I^2^ = 17.4%, *n* = 96) (Appendix A), and admission to the ICU (OR 1.32, 95% CI 1.16–1.49, I^2^ = 54.5%, *n* = 45) (Figure 4) among patients with diabetes compared to those who had no diabetes.

### 3.4. Chronic Respiratory Disease

Of the 559 included studies, 178 compared the odds of death vs. survival (*n* = 141), severe vs. moderate or mild illness (*n* = 40), and admission to the ICU vs. no admission (*n* = 27) among COVID-19 patients who had chronic respiratory disease vs. those without this problem. Of the 141 studies that compared the odds of death vs. survival, 66 report significantly greater odds of death among COVID-19 patients with chronic respiratory disease, and one reports significantly lower odds of death among COVID-19 patients with chronic respiratory disease; however, 75 document a lack of association between chronic respiratory disease and the odds of death. Out of the 40 studies that compared the odds of severe vs. mild or moderate illness among COVID-19 patients, 17 report significantly greater odds of severe illness, but 22 show a lack of association between chronic respiratory disease and the odds of developing severe illness. One study reports lower odds of severe illness among COVID-19 patients with chronic respiratory disease than among those without chronic respiratory disease. The association between having a chronic respiratory disease and the odds of admission to the ICU was assessed in 27 studies (nine reported increased odds, but one reported decreased odds). A meta-analysis of the 178 studies showed an association between chronic respiratory disease and increased odds of death (OR 1.55, 95% CI 1.42–1.68, I^2^ = 72.5%, *n* = 141) (Appendix A), severe illness (OR 1.37, 95% CI 1.07–1.66, I^2^ = 19.2%, *n* = 41) (Figure 5), and admission to the ICU (OR 1.27, 95% CI 1.01–1.53, I^2^ = 35.8%, *n* = 26) (Figure 6).

### 3.5. Cancer

A total of 127 studies examined the nature of the relationship between cancer and the odds of death (*n* = 86), severe illness (*n* = 35), and admission to the ICU (*n* = 22) among COVID-19 patients. Increased odds of death, severe illness, and admission to the ICU among COVID-19 patients who had cancer were reported in 31 (out of 86), 10 (out of 35), and 4 (out of 22) studies, respectively. One study showed decreased odds of death among COVID-19 patients who had cancer. The remaining studies showed a lack of association between having cancer and the odds of death (*n* = 84), severe illness (*n* = 25), and admission to the ICU (*n* = 18) among COVID-19 patients. A meta-analysis of the 127 studies showed increased odds of death among COVID-19 patients who had cancer (OR 1.55, 95% CI 1.34–1.76, I^2^ = 85.1%, *n* = 86) (Appendix A); however, there was a lack of correlation between this chronic disease and severe illness (OR 1.17, 95% CI 0.91–1.43, I^2^ = 15.6%, *n* = 35) (Appendix A) and admission to the ICU (OR 0.98, 95% CI 0.80–1.16, I^2^ = 0.0%, *n* = 22) (Appendix A) among COVID-19 patients.

### 3.6. Chronic Kidney Disease

A total of 161 studies examined the nature of the relationship between chronic kidney disease and the odds of death (*n* = 117), severe illness (*n* = 34), and admission to the ICU (*n* = 24) among COVID-19 patients. Of the 161 studies, 60 showed increased odds of death, 12 reported increased odds of severe illness, and 9 documented increased odds of admission to the ICU among COVID-19 patients who had chronic kidney disease vs. those who did not. One study reported decreased odds of death, two studies reported decreased odds of severe illness, and one showed decreased odds of admission to the ICU among COVID-19 patients with chronic kidney disease. The remaining studies showed a lack of association between chronic kidney disease and the odds of death (*n* = 56), severe illness (*n* = 20), and admission to the ICU (*n* = 14). A meta-analysis of the 161 studies showed an association between chronic kidney disease and increased odds of death (OR 1.73, 95% CI 1.54–1.92, I^2^ = 79.3%, *n* = 117) (Appendix A); however, there was a lack of correlation between chronic kidney disease and severe illness (OR 1.38, 95% CI 0.91–1.84, I^2^ = 75.3%, *n* = 34) (Appendix A) and admission to the ICU (OR 1.44, 95% CI 0.94–1.94, I^2^ = 53.1%, *n* = 24) among COVID-19 patients (Appendix A).

### 3.7. Chronic Liver Disease

Out of the 559 studies that are included in this review, 47 tested the association between chronic liver disease and death (*n* = 26), severe illness (*n* = 16), or admission to the ICU (*n* = 6) among COVID-19 patients. The majority of the studies report a lack of correlation between chronic liver disease and death (*n* = 20), severe illness (*n* = 14), and admission to the ICU (*n* = 6) among COVID-19 patients. Few studies report increased odds of death (*n* = 6) and severe illness (*n* = 2) among COVID-19 patients with chronic liver disease compared to those without this chronic health problem. A summary analysis of the 47 studies showed an association between chronic liver disease and increased odds of death (OR 1.50, 95% CI 1.31–1.68, I^2^ = 0.0%, *n* = 26); however, there was lack of correlation between chronic liver disease and severe illness (OR 0.98, 95% CI 0.67–1.30, I^2^ = 0.0%, *n* = 16) and admission to the ICU (OR 1.05 95% CI 0.62–1.48, I^2^ = 0.0%, *n* = 6) (Figure 7).

### 3.8. Cerebrovascular Diseases

A total of 54 studies report findings on the odds of death (*n* = 39), severe illness (*n* = 10), and admission to the ICU (*n* = 7) among COVID-19 patients with cerebrovascular diseases vs. those without this comorbidity. Out of 39 studies, 14 report increased odds of death, and one documents decreased odds of death among COVID-19 patients with cerebrovascular diseases. Some of these studies also show the association between cerebrovascular diseases and increased odds of developing severe illness (*n* = 5) and admission to the ICU (*n* = 4). The remaining 25 (out of 39), 5 (out of 10), and 3 (out of 7) of the studies reported that cerebrovascular disease is not associated with death, severe illness, and admission to the ICU in COVID-19 patients, respectively. A meta-analysis of the 54 studies showed increased odds of death (OR 1.59, 95% CI 1.24–1.93, I^2^ = 62.9%, *n* = 37) and severe illness (OR 1.89, 95% CI 1.25–2.53, I^2^ = 0.0%, *n* = 10) among COVID-19 patients with cerebrovascular diseases; however, the odds of admission to the ICU (OR 1.49, 95% CI 0.50–2.49, I^2^ = 14.2%, *n* = 7) were similar between patients with cerebrovascular diseases and those without this comorbidity (Figure 8).

### 3.9. Smoking

A total of 133 studies included in the current meta-analyses examined the impact of tobacco smoking on the clinical outcomes of COVID-19. Of these 133 studies, most showed a lack of association between smoking and death (68 out of 87), severe illness (35 out of 43), or admission to the ICU (13 out of 15) among COVID-19 patients. Only 17 studies show increased odds of death, 8 report increased odds of severe illness, and 2 report increased odds of admission to the ICU among COVID-19 patients who were former or current smokers compared to non-smokers. Two studies report decreased odds of death among COVID-19 patients who smoked cigarettes vs. those who did not. A meta-analysis of the studies showed increased odds of death (OR 1.22, 95% CI 1.01–1.43, I^2^ = 96.6%, *n* = 87) (Appendix A); however, the odds of severe illness (OR 1.06, 95% CI 0.09–1.22, I^2^ = 7.0%, *n* = 44) and admission to the ICU (OR 1.04, 95% CI 0.85–1.23, I^2^ = 27.6%, *n* = 14) were similar among COVID-19 patients who were former or current smokers and non-smokers (Figure 9). 

### 3.10. Age

A total of 241 studies that examined the association between age and COVID-19-related outcomes were included in this review. Out of 241 studies, 158 examined the effect of age on the odds of death, 74 assessed the effect of age on severe illness, and 30 examined the effect of age on admission to the ICU among COVID-19 patients. The studies treated age (in years) differently (≥60 vs. <60 (*n* = 68); ≥65 vs. <65 (*n* = 77); ≥50 vs. <50 (*n* = 6); ≥70 vs. <70 (*n* = 12); ≥75 vs. <75 (*n* = 7); >45 vs. ≤45 (*n* = 1)), and some treated age as a continuous variable (an increase by one year) (*n* = 70). Most of the studies that treated age as ≥65 vs. <65 years report increased odds of death (42 out of 44), severe illness (29 out of 32), and admission to the ICU (5 out of 6) among COVID-19 patients of ages ≥65 years, compared to those of <65 years. Similarly, the majority of the studies that treated age as ≥60 vs. <60 years report increased odds of death (39 out of 45), severe illness (16 out of 21), and admission to the ICU (7 out of 9) among COVID-19 patients of ages ≥60 years compared to those <60 years. Almost all of the studies that treated age as a continuous variable report increased odds of death (51 out of 52), severe illness (14 out of 14), and admission to the ICU (11 out of 11) with an increase in the age of COVID-19 patients by one year. The majority of the studies that treated age as ≥50 vs. <50 years, ≥70 vs. <70 years, and ≥75 vs. <75 years report increased odds of death (*n* = 2, 4, and 3, respectively) and severe illness (2, 1, and 2, respectively) among COVID-19 patients of ages ≥50 years, ≥70 years, and ≥75 years, compared to those <50 years, <70 years, and <75 years, respectively. A meta-analysis of these studies showed greater odds of death (OR 3.51, 95% CI 2.76–4.26, I^2^ = 96.7%, *n* = 44) (Appendix A), severe illness (OR 2.63, 95% CI 2.08–3.18, I^2^ = 69.5%, *n* = 32) (Appendix A), and admission to the ICU (OR 1.87, 95% CI 1.08–2.66, I^2^ = 79.7%, *n* = 6) (Appendix A) among patients of ages ≥65 years compared to those <65 years. The summary odds-ratio estimates of death (OR 4.32, 95% CI 3.39–5.25, I^2^ = 95.7%, *n* = 45) (Appendix A), severe illness (OR 2.14, 95% CI 1.63–2.66, I^2^ = 69.5%, *n* = 21)(Appendix A), and admission to the ICU (OR 1.92, 95% CI 1.51–2.33, I^2^ = 30.6%, *n* = 9) (Appendix A) among older-aged COVID-19 patients compared to younger ones were even much greater when the age was treated as ≥60 vs. <60. The increases in the odds of death (OR 1.06, 95% CI 1.05–1.07, I^2^ = 77.8%, *n* = 70), severe illness (OR 1.06, 95% CI 1.04–1.08, I^2^ = 74.7%, *n* = 14), and admission to the ICU (OR 1.03, 95% CI 1.01–1.05, I^2^ = 74.1%, *n* = 11) were also significant with an increase in the age of the COVID-19 patients by one year (Appendix A). COVID-19 patients of ages ≥75 years were also more likely to die than those with ages younger than 75 years (OR 2.40, 95% CI 2.09–2.70, I^2^ = 0.0%, *n* = 5). However, the odds of death, severe illness, and admission to the ICU among COVID-19 patients were comparable between individuals with ages ≥50 and <50 years, as well as between those with ages ≥70 and <70 years (Appendix A).

### 3.11. Gender

A total of 411 studies examined the effect of gender on the odds of death (*n* = 272), severe illness (*n* = 136), and admission to the ICU (*n* = 52) among COVID-19 patients. Of the 272 studies that compared the odds of death between males and females, 100 reported increased odds in males, but 3 showed decreased odds in males, and 159 documented similar odds between males and females. A meta-analysis of the 272 studies showed increased odds of death among males compared to females (OR 1.33, 95% CI 1.26–1.39, I^2^ = 89.3%) (Appendix A). Similarly, of the 136 studies that compared the odds of severe illness between males and females, 41 reported increased odds, 3 reported decreased odds, and 91 studies documented similar odds between males and females. A summary analysis of the 135 studies showed increased odds of severe illness among males compared to females (OR 1.26, 95% CI 1.16–1.37, I^2^ = 65.2%) (Appendix A); however, this could not be further assessed by tandem risk factors (e.g., underlying cardiovascular disease) in the available data.

A total of 52 studies examined the relationship between gender and the odds of admission to the ICU. Out of these 52 studies, 18 showed increased odds of admission to the ICU, and one reported decreased odds of admission to the ICU in males compared to females; however, 33 studies showed a lack of association between gender and the odds of admission to the ICU. A summary of the 52 studies showed increased admission to the ICU in males compared to females among COVID-19 patients (OR 1.40, 95% CI 1.24–1.55, I^2^ = 70.0%) (Appendix A).

### 3.12. Heterogeneity Assessment

There was no heterogeneity (I^2^ = 0.0%) observed among the studies included in the meta-analyses that estimated the summary odds ratios of death, severe illness, or admission to the ICU among COVID-19 patients with chronic liver disease vs. those without this comorbidity. There was also no or low heterogeneity among the studies included in the meta-analyses that examined the correlations between the specific clinical outcomes in COVID-19 patients and cardiovascular disease (I^2^ = 20.5% for severe illness), diabetes (I^2^ = 17.4% for severe illness), chronic respiratory disease (I^2^ = 19.2% for severe illness, I^2^ = 35.8% for admission to the ICU), cancer (I^2^ = 15.6% for severe illness, I^2^ = 0.0% for admission to the ICU), cerebrovascular disease (I^2^ = 0.0% for severe illness, I^2^ = 14.2% for admission to the ICU), smoking (I^2^ = 7.0% for severe illness, I^2^ = 27.6% for admission to the ICU), and age in years ≥60 vs. <60 (I^2^ = 30.6% for admission to the ICU), or ≥75 vs. <75 (I^2^ = 0.0% death and severe illness). There was moderate heterogeneity among the studies included in the meta-analyses that examined the correlation between admission to the ICU in COVID-19 patients and cardiovascular disease (I^2^ = 48.5%), diabetes (I^2^ = 54.5%), and chronic kidney disease (I^2^ = 53.1%). The heterogeneity level in the meta-analyses that was performed to examine the associations between hypertension and the odds of severe illness among COVID-19 patients was also moderate (I^2^ = 47.9%). 

However, the heterogeneity was high (I^2^ > 50%) among the studies that were combined in the meta-analyses to examine the associations between: (i) death and hypertension, cardiovascular disease, diabetes, cancer, chronic kidney disease, smoking, age ≥65 vs. <65 years, age ≥60 vs. <60 years, an age increase of one year, and gender in COVID-19 patients (Appendix A); (ii) severe illness and age, gender, and chronic kidney disease (Appendix A); and (iii) admission to the ICU and hypertension, diabetes, chronic kidney disease, and smoking (Appendix A). Subgroup analyses by study design, study area/country, sample size, year of publication, and income group status decreased the heterogeneity of the studies that assessed the relationship between comorbidities and demographic status and death (Appendix A), severe illness (Appendix A), and admission to the ICU (Appendix A) among COVID-19 patients. 

The meta-regression analysis also showed that the study area or country significantly affects the log ORs of death among male (vs. female) COVID-19 patients (meta-regression coefficient (β) = −0.50, *p* = 0.073) with diabetes (vs. without diabetes) (β = −0.23, *p* = 0.400). Similarly, the sample size significantly affected the log ORs of death among COVID-19 patients with ages ≥60 vs. <60 years (β = 8.74× 10^−6^, *p* = 0.631). The log odds of death among COVID-19 patients with chronic respiratory disease vs. those without this comorbidity were significantly different between studies published in 2020 and in 2021 (β = 8.74 × 10^−6^, *p* = 0.631). However, the meta-regression analysis showed a lack of effect of the study area or the country, the study design, the sample size, and the year of publication on the log ORs of death among COVID-19 patients with hypertension, cancer, cardiovascular disease, chronic kidney disease, those who smoke, and with ages ≥65 vs. <65 years (Appendix A). 

### 3.13. Publication Bias Assessment

The figures in File Appendix A display the funnel plots that were used to qualitatively examine the publication bias, and the meta-regression tests that were used to evaluate the asymmetry of the plots. The odds-ratio distributions for death vs. survival among COVID-19 patients with cancer vs. without cancer, with diabetes vs. without diabetes, with chronic liver disease vs. without chronic liver disease, with ages ≥60 vs. <60 years, and with ages ≥65 vs. <65 years, against their standard-error estimates and Egger tests for the asymmetry of the published articles, did not indicate publication bias. The funnel plots of the odds ratios of the likelihood of developing severe vs. moderate or mild illness, and the corresponding Egger tests, were also not significant among COVID-19 patients with hypertension, cancer, cerebrovascular disease, and chronic liver disease vs. those without these comorbidities, and vs. those who were smokers vs. nonsmokers. Studies that compared the odds ratios of admission to the ICU vs. no admission to the ICU among COVID-19 patients with cerebrovascular disease, chronic liver disease, and chronic kidney disease vs. those without these comorbidities, smokers vs. nonsmokers, and ages ≥65 vs. <65 years were also spread evenly on both sides of the average OR estimates, which created an approximately symmetrical funnel-shaped distribution. 

However, there was publication bias (i.e., the odds-ratio estimates were scattered asymmetrically in the funnel plot) among the studies that compared the odds of: (i) death vs. survival in patients with hypertension, obesity, cardiovascular disease, chronic respiratory disease, cerebrovascular disease, chronic kidney disease vs. those without the corresponding comorbidities; smokers vs. nonsmokers; males vs. females; and an age increase of one year; (ii) severe vs. mild or moderate illness in patients with diabetes, cardiovascular disease, chronic respiratory disease, chronic kidney disease vs. those without the corresponding comorbidities; males vs. females; ages ≥60 vs. <60 years; and ages ≥65 vs. <65 years; and (iii) admission to the ICU vs. no admission in patients with hypertension, cancer, diabetes, cardiovascular disease, and chronic respiratory disease vs. those without the corresponding comorbidities; males vs. females; ages ≥60 vs. <60 years; and an age increase of one year. 

### 3.14. Risk of Bias and Quality of the Studies

The risk of bias and the quality of the studies included in this systematic review are summarized in the Appendix A. Out of the 559 studies, 29 were good quality, 489 were moderate quality, and 45 were poor quality in terms of recruiting the study participants. The quality and bias were not assessed for nine studies. The majority of the studies also used good- (*n* = 397) or moderate-quality (*n* = 128) reliable and valid tools to determine the COVID-19 severity status and the related deaths among the study participants. In many studies (*n* = 310), the researchers or data collectors were not aware of the group (exposed vs. unexposed) to which the study participants belonged in the data-collection process and/or the participants were blinded to the research question. The majority of the included studies were of moderate quality in terms of the study design (retrospective case–control or cohort) (~90.0%), and of the controlling confounders (~70%) that may affect the relationship between comorbidities and COVID-19-related outcomes or death.

On the other hand, several studies were rated as low quality on the basis of the study design (*n* = 57), the data collection (*n* = 38), the confounders (*n* = 166), the blinding (*n* = 215), and the dropouts and withdrawals (*n* = 263). The total rating using the six criteria showed that 107 studies were of strong quality, 227 studies were of moderate quality, and 229 studies were of low quality. Studies were included in this review, regardless of their qualities.

## 4. Discussion

This systematic review and its meta-analyses confirm several correlations between demographic factors and comorbidities and severe illness and death among COVID-19 patients that have been reported in other reviews [5,6,7,8,9,10,11,12,13,14,15,16,17,18,19]. COVID-19 patients who were smokers, males, with ages ≥60 or 65 years, and those who had hypertension, diabetes, cardiovascular disease, cancer, chronic respiratory disease, chronic kidney disease, chronic liver disease, and cerebrovascular diseases were found to be more susceptible to death. The risk of developing severe illness also increased among male COVID-19 patients of ages 60 or 65 years, and among those who had hypertension, diabetes, chronic respiratory disease, cerebrovascular disease, and cardiovascular diseases. Comorbidities, including hypertension, diabetes, chronic respiratory disease, male sex, and older ages, were also associated with increased odds of admission to the ICU among COVID-19 patients.

Chronic diseases, including hypertension, diabetes, chronic respiratory disease, cardiovascular disease, chronic liver disease, and cerebrovascular diseases have myriad pathophysiologic impacts that are relevant to the outcomes from an infectious disease. For example, autoimmunity in diabetes may release inflammatory cytokines, such as IL-1β and TNFα, which contribute to a chronic inflammatory state [599]. Metabolic disorders may also impair the macrophage and lymphocyte function, which leads to low immune function [221], while the functional immunosuppression of senescence may play a role in older adults who experience worse disease [600]. Moreover, older adults also are more likely to have an underlying chronic disease [601]. The gender impacts on sepsis outcomes remain poorly characterized. 

The comorbidity and demography risk factors for death, severe illness, and admission to the ICU among COVID-19 patients may shift as the pandemic continues. SARS-CoV-2 changes rapidly through point mutations and recombination, and especially in its Spike and nucleocapsid regions [602,603]. This rapid ongoing evolution can alter the transmissibility and pathogenicity of the virus, and it may produce new variants that escape the host immune responses and that lead to more severe and fatal disease outcomes [604,605,606]. SARS-CoV-2 could also evolve to become less pathogenic [607]. The results of these interplays can be hard to predict. For instance, while the SARS-CoV-2 Omicron variant was less likely to cause severe illness in any given individual, its effective transmission resulted in high total case numbers and, thus, to high total severe disease burden and substantial health-system impacts in some locations. The very nature of the disease may also change. The SARS-CoV-2 variant A.30 has demonstrated tropism that is not observed for other viral variants and that could promote extrapulmonary spread and enhance the evasion from neutralization by antibodies [608].

The lack of association between some chronic diseases (e.g., kidney disease, chronic liver disease, and cancer) and smoking and severe illness or admission to the ICU in COVID-19 patients could be due to limitations in the original studies. The role of chronic kidney disease was assessed among comparatively few participants, which could explain its discordance with the results among its cardiovascular equivalents. Potential ascertainment bias in the original studies could also partly explain the lack of association between kidney disease, chronic liver disease, cancer, and smoking and severe illness or admission to the ICU, although these diseases are shown here and elsewhere to play a role in death. Similarly, the lack of correlation between smoking and severe disease in COVID-19 patients could be due to the watering effect of the chronic disease impacts on the outcomes, and particularly in older patients. If most of the patients with cardiopulmonary diseases were smokers, and if the results were exacerbated by ascertainment bias (if the care providers did not ask tobacco status enough, or misinterpreted prior smoking and current smoking as never smoking), then studies could miss the distinctive impacts of tobacco use. Unfortunately, most of the studies did not control for or explore the potential interactions and the confounding among chronic diseases, or smoking on COVID-19-related outcomes.

These findings have a number of public health and research implications. Better standardization of the definitions of comorbidities and the other risk factors in COVID-19 and other sepsis research would help to mitigate the heterogeneity that was observed in our analyses. Nonetheless, that older and otherwise medically vulnerable patients experience more severe COVID-19 outcomes is reinforced by this work. There are steps that clinicians and assistive care settings can now take. For example, the creation of awareness and the provision of robust infection prevention and control practices are indicated in high-risk patient settings, in and out of the hospital. Second, the higher likelihood of poor outcomes indicates a benefit from more intensive surveillance, patient monitoring, and early medical intervention in such patients. These findings support both early interruptive therapy and vaccine-boosting recommendations, although they suggest that further exploration is needed in the potential stratification within risk groups among those with hypertension, cardiovascular disease, diabetes, respiratory system disease, liver disease, and cerebrovascular disease. Additional prospective cohort studies are indicated to include studies that might look more closely at the unmet risk aspects, such as whether and how comorbidities and demographics can affect the risk of the acquisition of the SARS-CoV-2 relevant to the subsequent clinical outcomes of the disease. In addition, whether the stage or life expectancy confounds the results among cancer patients is not known.

There was a high level of heterogeneity and bias among the studies, and particularly among those that examined the effects of comorbidities and demographic factors on COVID-19-related death. The major sources of heterogeneity were the study design, the study country (including whether the country was considered middle or high income), the sample size, and the year of publication. Although the majority of the studies included were published in 2020, were retrospective in design, and were conducted in China, Italy, and the United States, several studies were cross-sectional or prospective, were conducted in other regions of the world, and were published in 2021. In addition, the sample sizes of the studies varied significantly, from <100 to >100,000. Moreover, the pathogens could be contributing to the heterogeneity, as different strains circulate within and between regions at different times, which potentially affects the observed COVID-19-related outcomes [607]. A subgroup analysis by the study design, the study area, and the sample size significantly decreased the heterogeneity.

This systematic review has several strengths. Unlike previous meta-analyses on related topics [5,6,7,8,9,10,11,12,13,14,15,16,17,18,19,20,21,22,23,24,25,26,27,28,29,30,31,32], this study: (i) involves a comprehensive analysis of a large number of studies that are based in different countries in the world; (ii) includes many articles that were published in 2021, after the introduction of the Delta strain; (iii) analyzes only peer-reviewed data; (iv) assesses the extensive analysis of the source of the heterogeneity and bias in the studies; and (v) compares the risk of admission to the ICU vs. no admission among hospitalized COVID-19 patients, and is stratified on the basis of comorbidities and demographic status.

Publication bias remains a challenge to a better understanding of the association between comorbidities and severe illness and death in COVID-19 patients. Most studies report frequencies or crude estimates that were not adjusted for potential confounders, which could affect the relationship between the comorbidities and the risk of severe illness or death in COVID-19 patients. In addition, the sample size and/or the number of severe cases or deaths in patients with varied comorbidities or demographic statuses were small in some studies, which increased the confidence interval estimates for the OR and decreased the power to reject false associations. These limitations in the original studies could have resulted in under- or overestimations of the summary OR estimates in the examination of the relationship between the demographics and comorbidities and this review’s outcomes. Moreover, the incorporation of a random-effects analysis does not fully mitigate the challenges from the cross-sectional and retrospective designs in the original studies to include the potential for underappreciated selection biases. The studies assessed and reported the exposures to cardiovascular diseases, diabetes, and cerebrovascular diseases separately and differently, and so composite analyses that explore the misclassification biases and the missed interactions could not be undertaken. There is a risk that some patients may have been represented in more than one study [609]. Furthermore, some studies may have reported biased results that are due to a conflict of interest with the organization that was sponsoring the research. For example, a published article that claimed that smokers are protected against SARS-CoV-2 infection was retracted because of relationships with the tobacco industry [610].

## 5. Conclusions

While older ages and chronic diseases were shown to increase the risk of developing severe illness, admission to the ICU, and death among the COVID-19 patients in our analysis, a marked heterogeneity was present when linking the risks with the outcomes. The effect of liver disease was a notable exception; while the data were limited, they were consistently associated with worse outcomes. The COVID-19 outcomes in women need further exploration. Standardized approaches to the collection of and the reporting on noncommunicable-disease risk factors and clinical outcomes would assist the recognition, prioritization, and development of the strategies against poor SARS-CoV-2 outcomes.

## Figures and Tables

**Figure 1 pathogens-11-00563-f001:**
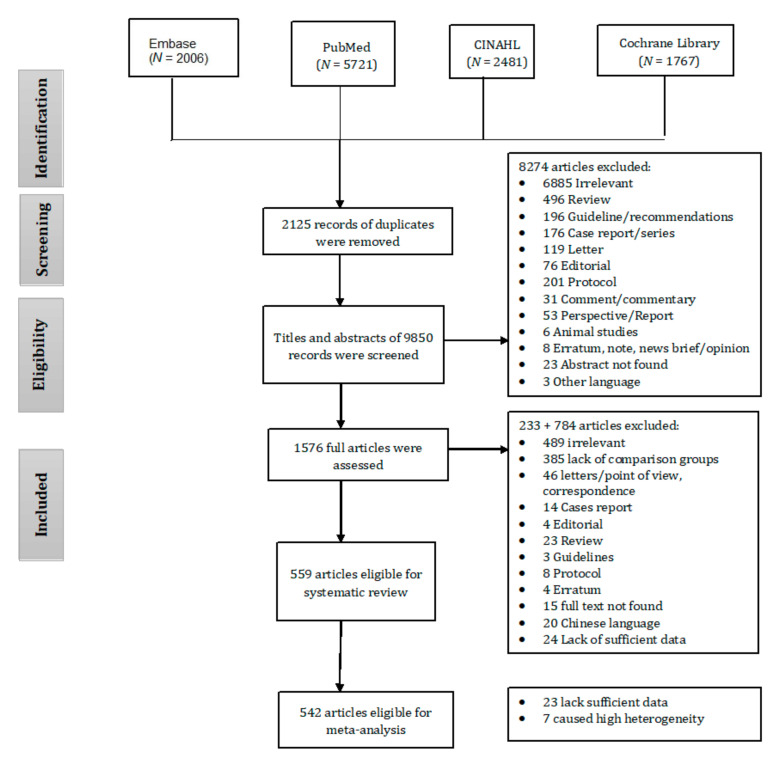
PRISMA flow diagram. Numbers of articles retrieved from databases, screened, excluded, and included.

**Figure 2 pathogens-11-00563-f002:**
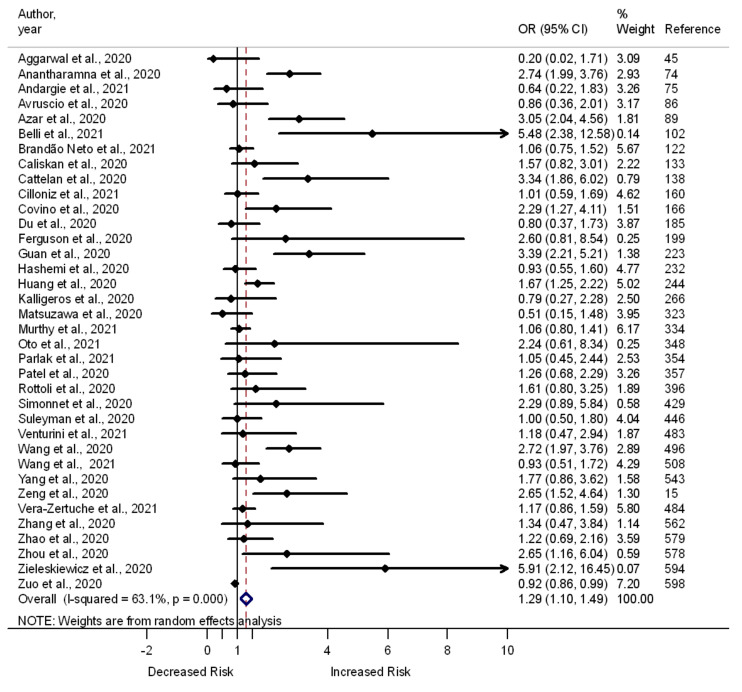
Forest plot showing the relationship of hypertension with the odds of admission to the ICU among COVID-19 patients.

**Figure 3 pathogens-11-00563-f003:**
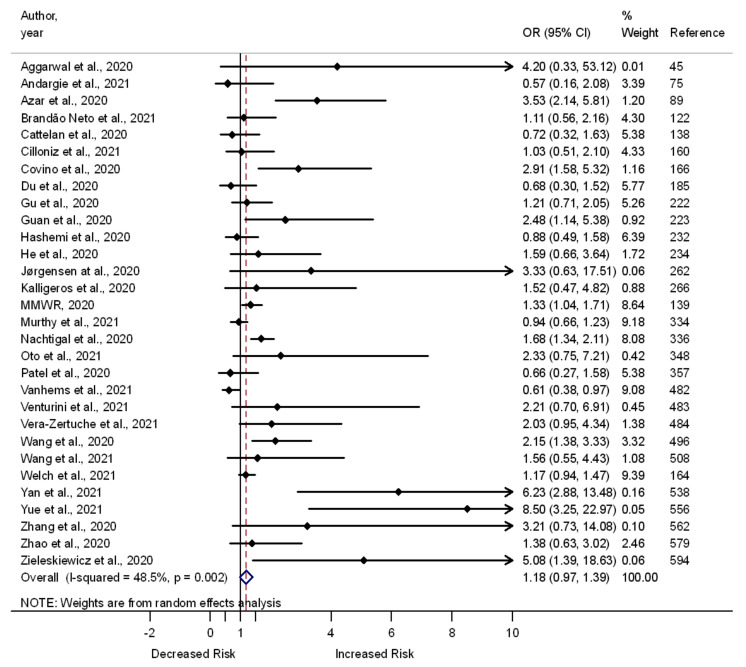
Forest plot showing the relationship of cardiovascular disease with the odds of admission to the ICU among COVID-19 patients.

**Figure 4 pathogens-11-00563-f004:**
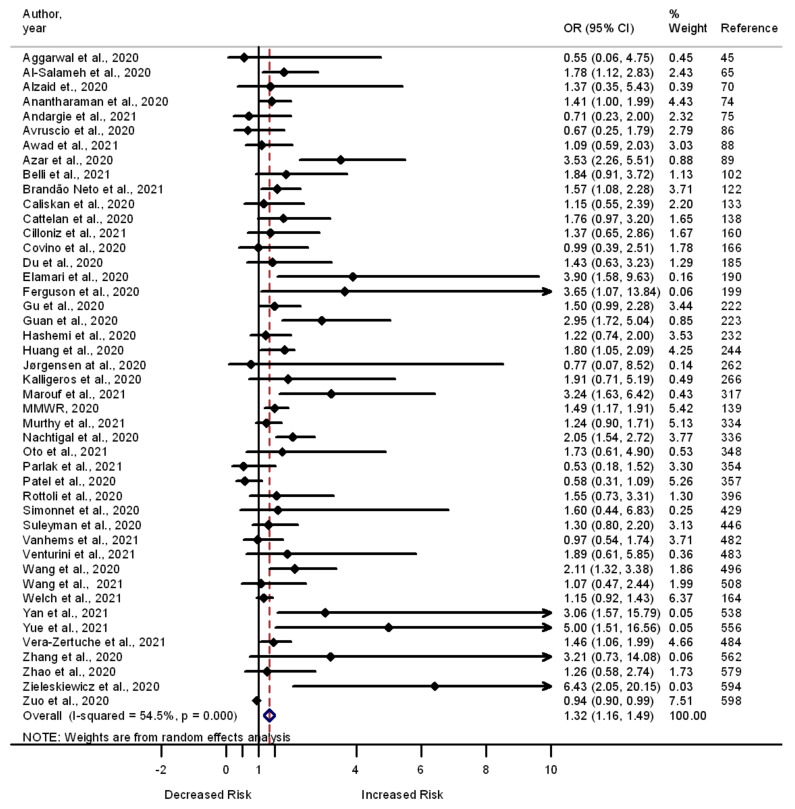
Forest plot showing the relationship of diabetes with the odds of admission to the ICU among COVID-19 patients.

**Figure 5 pathogens-11-00563-f005:**
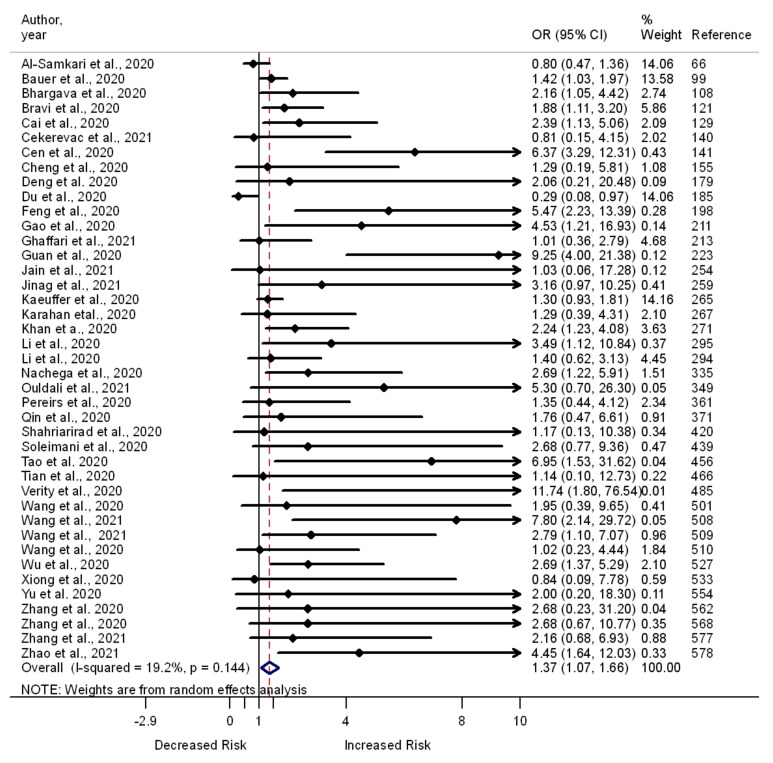
Forest plot showing the relationship of chronic respiratory diseases with the odds of severe illness among COVID-19 patients.

**Figure 6 pathogens-11-00563-f006:**
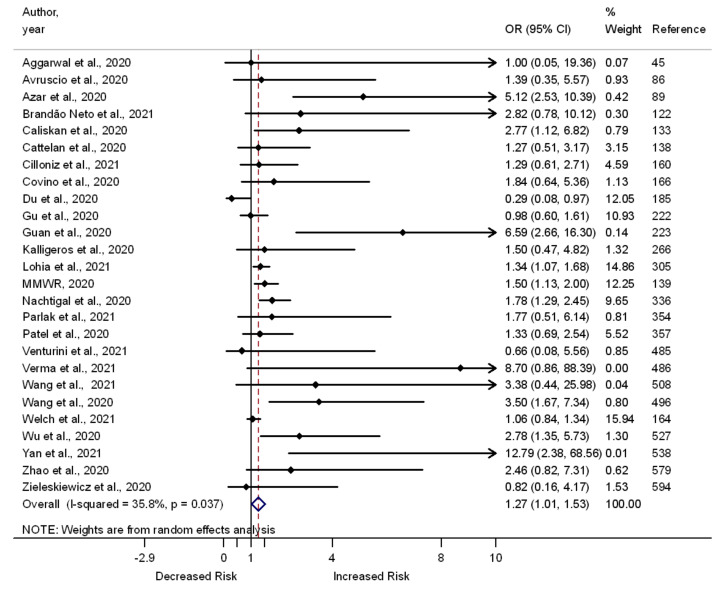
Forest plot showing the relationship of chronic respiratory diseases with the odds of admission to the ICU among COVID-19 patients.

**Figure 7 pathogens-11-00563-f007:**
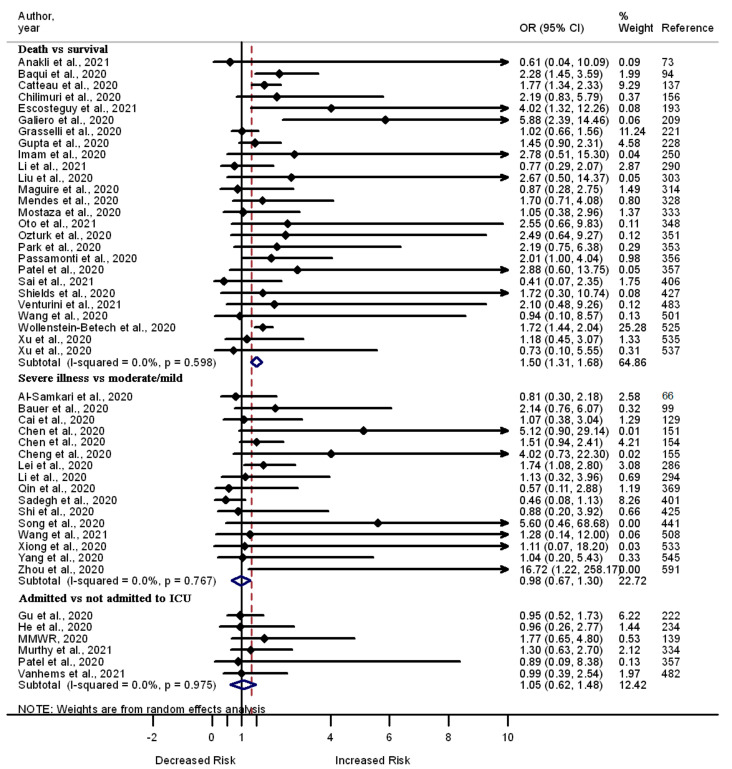
Forest plot showing the relationship between chronic liver disease and the odds of death, severe illness, and admission to the ICU among COVID-19 patients.

**Figure 8 pathogens-11-00563-f008:**
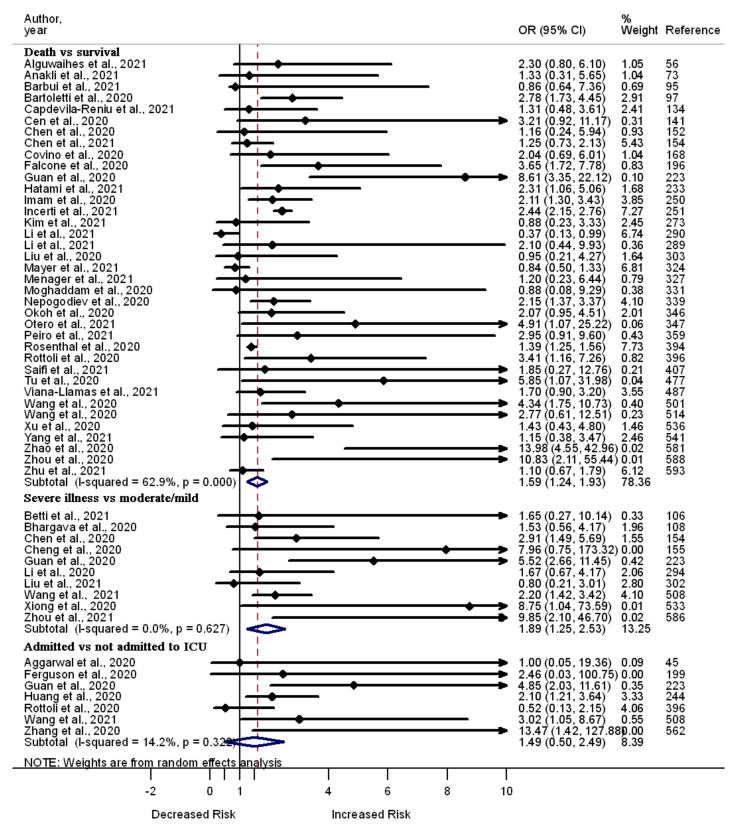
Forest plot showing the relationship of cerebrovascular diseases with the odds of death, severe illness, and admission to the ICU among COVID-19 patients.

**Figure 9 pathogens-11-00563-f009:**
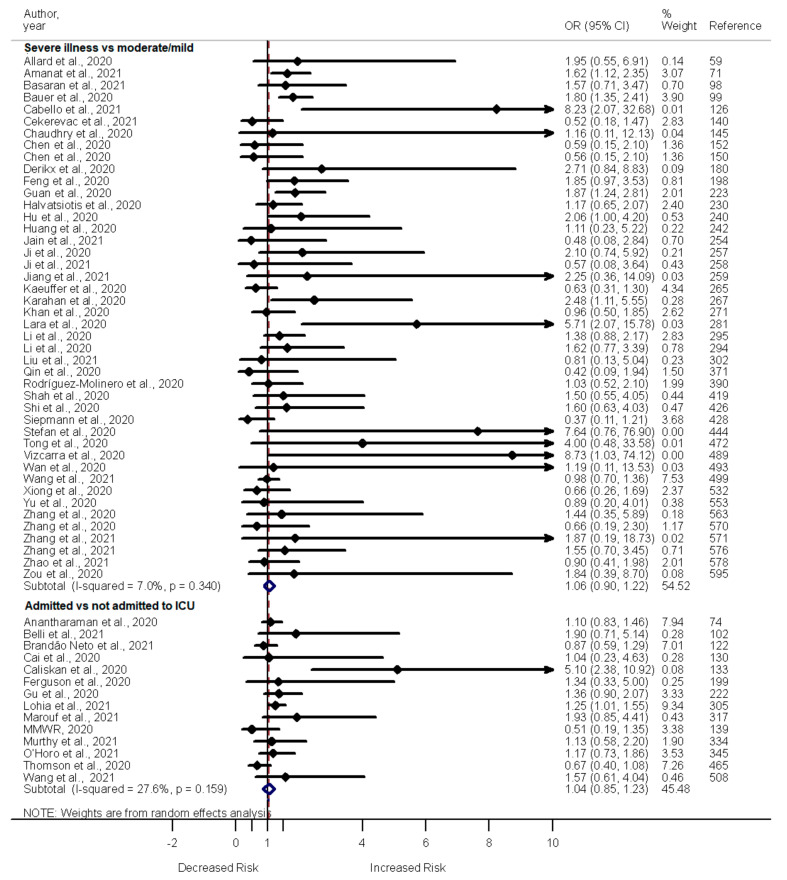
Forest plot showing the relationship of smoking with the odds of severe illness and admission to the ICU among COVID-19 patients.

## Data Availability

All data generated or analyzed during this study are included in this published article and in its Appendix A.

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
