# Peer review of "Heterogeneity and Risk of Bias in Studies Examining Risk Factors for Severe Illness and Death in COVID-19: A Systematic Review and Meta-Analysis"

_pathogens, 2022, doi:10.3390/pathogens11050563_

Round 1

Reviewer 1 Report

The submitted paper conducted a meta-analysis of COVID risk factors and found that comorbidities, male sex, and age were substantially associated with COVID-19-related severe illness, ICU hospitalization, and mortality. However, the conclusion's robustness is compromised by the quality of publications and the substantial variability of pooled estimates. Nevertheless, such issues are unavoidable when conducting meta analyses on hundreds of papers. In general, this work is of exceptional quality. Congratulations for a job well done.

Author Response

We thank the reviewer for the strong evaluation. We acknowledge the reviewer's note that the high heterogeneity among the studies included in this review could have affected the final conclusions. So, we conducted a comprehensive analysis of the sources of heterogeneity by applying subgroup analyses and meta-regression tests. We have also discussed heterogeneity as a limitation of this meta-analysis study.

Reviewer 2 Report

Major comments:

It is a very interesting article. The authors performed a systematic review and meta-analysis of almost 12,000 articles and identified 542 studies about the impact of demographics and co-morbidities on clinical outcome of COVID-19. Their meta-analysis differs from previous ones because it tried to address severe limitations and weaknesses of previous studies. They observed a significant heterogeneity in linking risks with outcomes. The analysis was performed carefully and the manuscript was well written and easy to follow, with some exceptions (some rather tiring parts that could be summarized) in the results (see below). The authors also clearly discuss the strengths and weaknesses of their approach.

It is possible that such heterogeneity is partly due to the quality of the various healthcare systems. It would be interesting to run the same analyses for one group of studies from lower-income countries vs higher income countries.

Line 55: the authors have analyzed relevant literature that was published before May 2021. It would be useful here to provide another updated number, on how many new studies have been published until April 2022, without of course incorporating them in their analyses. Just to understand how much new information has been gathered in the meantime. However, this is not mandatory, if it constitutes a great amount of work for the authors.

Line 3.9: Smoking. This is a very important and controversial section, because to my knowledge, people who have obtained funding from tobacco companies have published dubious papers about the protective effect of smoking! And they have not stated their conflict of interest, resulting in retracted papers. I would strongly recommend the authors to include a forest plot about smoking too and also discuss that papers have been retracted (somewhere within lines 551-558). See:

https://www.theguardian.com/science/2021/apr/22/scientific-paper-claiming-smokers-less-likely-to-acquire-covid-retracted-over-tobacco-industry-links

Section 3.12: The way that the results are presented is rather tiring. Could the authors somehow summarize all these in a figure or table and briefly discuss them in the paper, while putting this entire section in the supplementary material? Same for section 3.13.

Lines 517-521: Given these levels of quality among the various studies, maybe the various forest plots could incorporate which studies were of high/moderate/low quality.

In the Discussion, I would strongly recommend the authors to include a short section/paragraph about the evolution of Coronaviruses and especially SARS-CoV-2. This is a family of viruses that evolves very rapidly by both point mutations and recombination, especially at its Spike region (see: doi: 10.1093/molbev/msab292. and https://doi.org/10.3390/v14040707). This rapid evolution is very important because the Spike can affect the pathogenicity of the virus. The emergence of the Omicron and Omicron 2 variants of concern is the biggest proof, where the pathogenicity of the virus and its tissue tropism (towards the upper respiratory) have changed. Such mutations especially in the Spike may alter the properties and pathogenicity of the virus. For example, the A.30 SARS-CoV-2 lineage has demonstrated enhanced evasion from vaccine-induced antibodies and altered cell-entry properties with a preference for other cell-types that would promote extra-pulmonary spread (https://doi.org/10.1038/s41423-021-00779-5). In another example from another cat coronavirus, point mutations in the Spike are sufficient to transform a benign strain into a deadly strain (DOI: 10.1128/JVI.79.22.14122-14130.2005). Therefore, it is possible that part of this heterogeneity in the various studies could also be due to different strains circulating. In addition, this rapid evolution of the Spike means that many of these risk factors identified by previous studies may change in the future, as the pandemic continues. I believe that all of the above would greatly benefit the paper, if they are discussed.

Minor comments:

Line 28: I would remove the million after the number.

Line 73: in vitro: please italicize.

Figure 2: is the Kalligeros et al a duplicate?

Figure 7. one weight (Zhou Yiwu) is not clear.

Figure 8: Please check it.

Line 404: Italy

Line 414: year

Line 562: ill patient?

Author Response

Dear Dr.

We would like to thank you for handling the submission and review process. We also thank the reviewers for their careful review and constructive comments, which have helped to improve the manuscript. We have made changes to the manuscript based on reviewers' suggestions and describe these changes in the below paragraphs. We hope that you will find our responses acceptable, and we look forward to your decision.

Reviewer 2

Major comments:

It is a very interesting article. The authors performed a systematic review and meta-analysis of almost 12,000 articles and identified 542 studies about the impact of demographics and comorbidities on the clinical outcome of COVID-19. Their meta-analysis differs from previous ones because it tried to address severe limitations and weaknesses of previous studies. They observed significant heterogeneity in linking risks with outcomes. The analysis was performed carefully and the manuscript was well written and easy to follow, with some exceptions (some rather tiring parts that could be summarized) in the results (see below). The authors also clearly discuss the strengths and weaknesses of their approach. 

  1. It is possible that such heterogeneity is partly due to the quality of the various healthcare systems. It would be interesting to run the same analyses for one group of studies from lower-income countries vs higher income countries.

Response:  We have conducted a subgroup analysis based on the income group status and summarized the heterogeneity results in supplementary tables 19, 20 & 21 and briefly discussed it in the results' Heterogeneity assessment". There were few studies conducted in low-income countries included in this review. So, heterogeneity results were only for middle and high-income countries.

  1. Line 55: the authors have analyzed relevant literature that was published before May 2021. It would be useful here to provide another updated number, on how many new studies have been published until April 2022, without of course, incorporating them in their analyses. Just to understand how much new information has been gathered in the meantime. However, this is not mandatory if it constitutes a great amount of work for the authors.

Response: We liked this idea and explored feasibility. We had employed a comprehensive search strategy when we did this work. Repeating it for this purpose, even restricting the articles to those published between May 18, 2021, and April 30, 2022, yielded 49,882 articles in PubMed, 72,944 in Embase, 3164 in Cochrane Library, and 10,852 in CINHAL. There certainly are many duplicates in this output and we suspect that the vast majority of the  125,990 results (PubMed + Embase + Cochrane Library + CINHAL) would not be eligible. In order to make such an estimate in the context of our work, however, we applying our rigorous screening process would not be feasible.

  1. Line 3.9: Smoking. This is a very important and controversial section, because to my knowledge, people who have obtained funding from tobacco companies have published dubious papers about the protective effect of smoking! And they have not stated their conflict of interest, resulting in retracted papers. I would strongly recommend the authors to include a forest plot about smoking too and also discuss that papers have been retracted (somewhere within lines 551-558). See:

https://www.theguardian.com/science/2021/apr/22/scientific-paper-claiming-smokers-less-likely-to-acquire-covid-retracted-over-tobacco-industry-links.

Response: We have included a forest plot (Fig 9) showing study findings on the odds ratio of admission to ICU among COVID-19 patients who were former or current smokers vs non-smokers as a main file of the manuscript. We have also discussed potential bias in the summary estimates due to links/conflicts of interest between the researchers and the industry. The added text in the Discussion reads, ' Furthermore, some studies may have reported biased results due to conflict of interest with the organization sponsoring the research. For example, a published article claiming that smokers are protected against SARS-CoV-2 infection was retracted due to relation-ships with the tobacco industry [53].” (See line 661 to 664).

  1. Section 3.12: The way that the results are presented is rather tiring. Could the authors somehow summarize all these in a figure or table and briefly discuss them in the paper, while putting this entire section in the supplementary material? Same for section 3.13.

Response: We have summarized results from heterogeneity analysis in supplementary tables 19, 20 & 21 and briefly discussed the results under the heading "Heterogeneity assessment." We have also summarized results from publication bias in supplementary Fig 7 and briefly discussed the findings in the paper under the heading "publication bias assessment."

  1. Lines 517-521: Given these levels of quality among the various studies, maybe the various forest plots could incorporate which studies were of high/moderate/low quality.

Response: We acknowledge the reviewer's observation that adding further detail about the quality of the studies in the forest plots would be useful to some readers. It also would make the plots more complex to understand for other readers or substantially add to the number of plots present, however, and the information is available elsewhere in the work. Quality is part of what influences factors leading to the I2 present in the forest plots, but fully appreciate that that does not roundly meet the reviewer’s point. Due to the volume of articles included in this review, to adjust the forest plot or generate quality weighted plots would require more time than the journal’s turnaround requirement for remitting this correction. Should the editorial team wish that this be pursued enroute publication we are very happy to undertake it if the revised timeline is conducive

  1. In the Discussion, I would strongly recommend the authors to include a short section/paragraph about the evolution of Coronaviruses and especially SARS-CoV-2. This family of viruses evolves very rapidly by both point mutations and recombination, especially at its Spike region (see: doi: 10.1093/molbev/msab292. and https://doi.org/10.3390/v14040707). This rapid evolution is very important because the Spike can affect the pathogenicity of the virus. The emergence of the Omicron and Omicron 2 variants of concern is the biggest proof, where the pathogenicity of the virus and its tissue tropism (towards the upper respiratory) have changed. Such mutations, especially in the Spike may alter the properties and pathogenicity of the virus. For example, the A.30 SARS-CoV-2 lineage has demonstrated enhanced evasion from vaccine-induced antibodies and altered cell-entry properties with a preference for other cell-types that would promote extra-pulmonary spread (https://doi.org/10.1038/s41423-021-00779-5). In another example from another cat coronavirus, point mutations in the Spike are sufficient to transform a benign strain into a deadly strain (DOI: 10.1128/JVI.79.22.14122-14130.2005). Therefore, it is possible that part of this heterogeneity in the various studies could also be due to different strains circulating. In addition, this rapid evolution of the Spike means that many of these risk factors identified by previous studies may change in the future, as the pandemic continues. I believe that all of the above would greatly benefit the paper, if they are discussed.

Response: We thank the reviewer for this suggestion. We have discussed how the evolution of SARS-CoV-2 plays a role in the findings of risk factors for severe illness, admission to ICU, and death in COVID-19 patients and the heterogeneity in the revised manuscript. The added text on the discussion readsApologies,

"Comorbidity and demography risk factors for death, severe illness, and admission to ICU among COVID-19 patients may shift as the pandemic continues. SARS-CoV-2 changes rapidly through point mutations and recombination, especially in its Spike and nucleocapsid regions [45,46]. This rapid ongoing evolution can alter the transmissibility and pathogenicity of the virus and may produce new variants that escape host immune responses and lead to more severe and fatal disease outcomes [47-49]. SARS-CoV-2 could also evolve to become less pathogenic [50]. The results of these interplays can be hard to predict. For instance, while the SARS-CoV-2 Omicron variant was less likely to cause severe illness in any given individual, its effective transmission resulted in high total case numbers and so high total severe disease burden and substantial health system impact in some locations. The very nature of the disease may also change. The SARS-CoV-2 variant A.30 has demonstrated tropism not observed for other viral variants that could promote extrapulmonary spread and enhance evasion from neutralization by antibodies [51]." (Line 578 to 591).

“Moreover, the pathogens could be contributing to heterogeneity as different strains circulate within and between regions at different times, potentially affecting observed COVID-19 related outcomes [50]. Subgroup analysis by the study design, study area, and sample size significantly decreased the heterogeneity. (Line 634 to 637).

  1. Minor comments:
  2. Line 28: I would remove the million after the number.

Response: Thank you, we have changed that syntax

  1. Line 73: in vitro: please italicize.

Response: 'in vitro' italicized in line 68

  1. Figure 2: is the Kalligeros et al a duplicate?

Response: we have removed the duplicate 'Kalligeros et al' and updated the figure in the revised MS.

  1. Figure 7. one weight (Zhou Yiwu) is not clear.

Response: we have made clear the weight for Zhou Yiwu and updated the figure in the revised MS.

  1. Figure 8: Please check it. Response. We have checked the figure.
  2. Line 404: Italy. Response. We have replaced the word 'Italia' with 'Italy' in (line 630)
  3. Line 414: year. We have replaced the word 'years' with 'year' (line 425)
  4. Line 562: ill patient? We have removed the phrase 'ill patients' from the revised MS. (line 610/611)